# Tribological Properties of Solid Solution Strengthened Laser Cladded NiCrBSi/WC-12Co Metal Matrix Composite Coatings

**DOI:** 10.3390/ma15010342

**Published:** 2022-01-04

**Authors:** Zoran Bergant, Barbara Šetina Batič, Imre Felde, Roman Šturm, Marko Sedlaček

**Affiliations:** 1Faculty of Mechanical Engineering, University of Ljubljana, Aškerčeva 6, 1000 Ljubljana, Slovenia; 2Institute of Metals and Technology Ljubljana, Lepi pot 11, 1000 Ljubljana, Slovenia; barbara.setina@imt.si (B.Š.B.); marko.sedlacek@imt.si (M.S.); 3John von Neumann Faculty of Informatics, Óbuda University, Bécsi út 96/B, 1034 Budapest, Hungary; felde.imre@uni-obuda.hu

**Keywords:** wear, solid-solution strengthening, laser cladding, NiCrBSi, WC-Co, coating

## Abstract

NiCrBSi, WC-12Co and NiCrBSi with 30, 40 and 50 wt.% WC-12Co coatings were produced on low carbon steel by laser cladding with an Nd:YAG laser with a multi-jet coaxial cladding-nozzle. The microstructure properties after WC-12Co alloying were investigated by scanning electron microscopy (SEM), energy dispersive spectroscopy (EDS), X-ray diffraction (XRD), electron backscatter diffraction (EBSD) and Vickers hardness tests. The resulting microstructures consisted of a γ-Ni and Ni_3_B matrix, strengthened with Co and W, Ni_3_Si, CrB, Cr_7_C3, Cr_23_C_6_, WC/W_2_C phases. In coatings with 30, 40 and 50 wt.% WC-12Co, a solid solution, strengthened multi-matrix NiCrWCo phase formed, which yielded a higher matrix hardness. Wear tests that monitored the friction coefficients were performed with a tribometer that contained a ball-on-disc configuration, Al_2_O_3_ counter-body and reciprocal sliding mode at room temperature. The major wear mode on the NiCrBSi coatings without the WC-12Co was adhesive with a high wear rate and visible material loss by flaking, delamination and micro-ploughing. The addition of WC-12Co to the NiCrBSi coating significantly increased the wear resistance and changed the major wear mechanism from adhesion to three-body abrasion and fatigue wear.

## 1. Introduction

Research of the interactions among different combinations of materials leads to advancements in the development of metal matrix composites (MMCs) for surface coating applications and 3D printing [1,2,3,4]. Laser cladding has been employed in many industries as a versatile additive technology for precision coating deposition [5,6] to improve the properties of low-alloyed substrates and for printed parts [7]. This process involves feeding a stream of metallic powder or wire into a melt pool that is generated by a laser beam. Hard metals or cemented carbides are composite materials with one or more hard and brittle carbide phases bonded together by a soft and ductile binder, such as Co, Ni and Fe.

NiCrBSi is a self-fluxing alloy with a low-temperature melting point that does not require the addition of fluxing elements to wet the substrate or reinforcing particles [8]. Boron (B) acts as a fluxing agent because it reduces the melting temperature of a Ni-based matrix by forming the Ni-Ni_3_B eutectic phase and increases the hardness with the formation of carbo-borides [9]. Silicon acts as a fluxing agent as it reduces the surface tension of the melt, which results in the improved wetting of the substrate surface and reinforcing particles [10]. The partially soluble alloying elements Cr, Si and B reduce the liquidus temperature of the coating [11]. NiCrBSi coatings have been successfully used to improve corrosion resistance, but they often do not have a high wear resistance, especially in comparison with ceramic materials [12]. In addition, adhesive wear of NiCrBSi coatings can lead to severe material removal despite having a relatively high hardness. However, the self-fluxing properties and low melting temperature of NiCrBSi alloys make them attractive for use as a primary binder to efficiently wet, adhere and bind the reinforcing particles. Numerous studies have been conducted to improve the wear resistance of this coating with the addition of hard carbides, such as WC [13], SiC [14], TiC [15], Cr_3_C_2_ [16], TiN [12] and Y_2_O_3_ [17].

One of the most versatile and widely used hard carbides is tungsten carbide (WC), which is typically used for cutting tool applications in the form of tungsten-carbide cobalt (WC-Co) and for coatings. Tungsten carbides have a high melting point of 2870 °C, high corrosion and wear resistance and good electrical and high thermal conductivity.

Electron backscatter diffraction (EBSD) is a tool that is mainly used to visualize crystallographic characteristics, such as the orientation, grain size and shape, grain boundary misorientation, crystallographic texture and orientation relationships between different phases. However, it has also been used to identify and distinguish different phases in many metallic and nonmetallic systems, especially in combination with elemental dispersive spectroscopy (EDS) compositional analysis with a high spatial resolution. By using this approach for the identification of phases, it is possible to distinguish between different types of carbide and boride precipitates based on their crystallographic properties. Hemmati et al. [18] studied NiCrBSiC coatings by combining XRD, EBSD and TEM analyses. They identified two types of boride precipitates, namely polycrystalline orthorhombic CrB and single-crystal tetragonal Cr_5_B_3_. The carbides comprised the Cr_7_C_3_ type, with a hexagonal crystal structure. In a recent publication, Chen et al. [19] studied the crystallographic texture of plasma-sprayed-remelted NiCrBSi coatings by EBSD and XRD. They identified the carbides in the coating as Cr_7_C_3_. There were also two types of borides, namely CrB and Cr_3_B_4_, in addition to grains of the Ni_3_B phase in the microstructure.

Sheppard et al. [20] investigated electrical-arc deposited NiCrBSi/WC coatings. They reported a substantial improvement in the wear resistance with the addition of WC. They also reported WC dissolution, leading to the formation of W_2_C particles, which detached from the contacting surface during wear and formed debris and craters on the surface. Zhou et al. [21] investigated the microstructures of laser induction hybrid cladded NiCrBSi/35%WC before and after WC decomposition and observed that herringbone and lamellar shaped carbides formed after interacting with the matrix. Although thermal spray processes, such as combustion, plasma, electric-arc, high-velocity or cold spraying, offer high production rates, they often have a low adhesion strength and a large number of pores [22].

Laser cladding processes are more suitable for a local and precise deposition that provides good metallurgical bonding. During the laser cladding process, the interaction and mixing of alloying elements occur in the molten pool due to Marangoni convection and buoyancy forces; therefore, the process is more suitable for the deposition of multi-matrix coatings than thermal spraying or cold-spraying processes. Fernández et al. [23] studied the wear behaviour of a laser cladded NiCrBSi alloy reinforced with different amounts of WC. They reported that the wear decreased exponentially with the actual percentage of WC. In addition, they noted that a significant wear decrease did not occur above the actual 30 wt.% WC on the surface. The main wear mechanisms were adhesive and oxidative on the pure NiCrBSi coatings, and abrasive three-body wear in the case of the NiCrBSi-WC coatings. They used pure WC, but this material can be pre-alloyed with Ni or Co to obtain complementary effects or similar particle sizes after alloying. A Ni addition can improve the WC-Co strength through solid solution strengthening [24]. With an increase in the Ni content, the transverse rupture strength and room temperature compression strength also increase [25]. Guo et al. [26] investigated the high-temperature wear resistance of laser cladded NiCrBSi WC pre-alloyed with Ni. This combination may form a more homogeneous composite alloy or a similar particle size as the mixing powder at the expense of a reduction in the overall carbide content.

Yao [27], Vencl et al. [28] and Bolelli et al. [29] studied high-velocity oxy-fuel deposited nickel-based coatings with the addition of WC pre-alloyed with cobalt. Cobalt, in this case, is a secondary binder and highly soluble in a Ni matrix. The advantage of using WC pre-alloyed with Co is that cobalt and nickel form a solid solution, which increases the strength and hardness and improves properties at elevated temperatures. During thermal spraying, the powder particles, also called splats, form a lamellar and heterogeneous microstructure without intensive alloying between the splats or substrate. Mixing and alloying of powder chemical elements can be achieved more efficiently in the molten pool during laser cladding due to Marangoni convection and buoyancy forces. Alloying the Ni-based matrix with cobalt reduces plastic deformation of the matrix between the hard particles, the friction when surfaces slide against each other and the adhesion wear. The metal binder is usually removed first, followed by the WC particles due to the weakened bonding, resulting in the removal of additional WC particles [27]. WC-Co is also used as an independent deposition material [30]; however, it has a small parametric window for successful cladding because it is susceptible to cracking/detaching/spalling and dissolution of the carbides accompanied by the formation of carbon-oxide gaseous porosity if overheated [31].

A known problem concerning the laser cladding of WC-Co systems is that intense heating of the WC leads to WC decomposition and precipitation of carbon in the form of graphite and later free carbon and wolfram [32]:2WC⇌W_2_C + C(1)
W_2_C→2W + C(2)

In general, reaction (1) is bidirectional, where WC forms from the W_2_C by slow diffusion [33]. During welding, tungsten carbide decomposes to ditungsten carbide W_2_C and carbon as graphite at approximately 1250 °C [34]. The typical temperature in a melt pool during normal laser cladding far exceeds 1250 °C; typical temperatures in the melt pool range from 1800–2300 °C. Furthermore, when graphite reacts with oxygen in the atmosphere, it forms CO and CO_2_ gas, which can contribute to heat input as an active gas. A decrease in the carbon in the matrix leads to the formation of brittle ternary eutectic phases of W, Co, and C and brittle particles, such as (Co_3_W_3_)C, which form when there is an abundance of cobalt [32]. The round shapes and small size of the carbides indicate that they dissolve partially due to the heat generated during their deposition. All the aspects above increase the difficulty of depositing pure WC-Co systems.

In this study, we characterized the laser cladded microstructures and wear properties of NiCrBSi and WC-12Co coatings with different mixing proportions. The novelty of this research is the in-depth analysis and discussion of the effects of the solid solution strengthening of the nickel matrix with the interaction of alloying elements in the laser melt pool on microstructure, hardness and corresponding wear mechanisms as a consequence of matrix strengthening, which leads to better protective properties of coatings. The combination of those two materials and binders, processed in the melt pool, reinforced and enriched the NiCrBSi coating and increased the wear resistance. We have shown that the contribution of hard WC can have a beneficial effect on wear rates, demonstrating that the simultaneous melting of both binders by the laser heat source led to a multi-matrix alloy with better properties than the original binders. Moreover, the combination of WC-Co with a Ni-self-fluxing alloy decreased the exposure of WC-Co to the direct laser beam. Thus, adding WC pre-alloyed with cobalt into NiCrBSi had the complementary effect of strengthening and increasing the toughness of the original Ni-matrix binder of the self-fluxing alloy. The research contributes to the knowledge of microstructural evolution and phases formed as well as the unlubricated wear behaviour of different NiCrBSi-WC-Co matrix systems after laser cladding. Findings and data can be useful in designing new industrial coating applications using similar metallic/ceramic matrix coating systems.

## 2. Materials and Methods

### 2.1. Substrate and Coating Materials

The coatings were laser cladded on a substrate made from low-carbon (0.08% C) mild steel (W. No. 1.0037, EN 10027-2) with dimensions of 50 × 25 × 10 mm^3^. The surface of the steel substrate was milled to obtain an average initial roughness of *S_a_* = 1.1 μm and degreased with ethanol. The NiCrBSi powder produced with gas atomization from Castolin Eutectic, referenced as grade Eutalloy RW, was chosen as the primary feedstock material, as shown in Figure 1a. The chemical composition of the Eutalloy RW 12495 feedstock powder material is given in Table 1. The nickel-based powder contained carbide-forming alloying elements Cr and B, which increased wear resistance. In contrast, the presence of Si increased the wettability and allowed deoxidation reactions to occur in the coating during the heat treatment. The NiCrBSi alloy had the lowest liquidus temperature of 1110 °C, which was far below the liquidus temperature of the low-carbon steel substrate, which was approximately 1500 °C. The NiCrBSi eutalloy RW 12495 particle size was normally distributed with a mean value of 72 μm [35].

The agglomerated WC-12 wt.% Co cermet Sulzer Metco 5810 was used as a secondary powder and mixed with the NiCrBSi powder, as shown in Table 2. In Figure 1b, a mixed NiCrBSi + 30 wt.% WC-Co powder is shown and in Figure 1c, the WC-12Co powder is shown. The coating resulting from the addition of WC-12Co had good resistance to sliding wear, impact, abrasion and can be used for the coating of steel rolls, sink rolls, exhaust fans, pump housings, conveyor screws and rod couplings up to 500 °C, for example. The average particle size ratio of the mixed powders was ~1.14, similar to the size of the NiCrBSi powder, but the difference was not too large to produce transport segregation during feeding. Cobalt is a binder responsible for densification through the wetting, spreading and formation of agglomerates during liquid-phase sintering [36].

### 2.2. Laser Cladding Process

One-step powder laser cladding with a multi-jet coaxial nozzle was used to deposit pre-selected powder alloys. This process enables deposition and remelting to occur almost simultaneously and represents an efficient and practical cladding method with minimal thermal effects for the substrate/base material [37]. A schematic of the multi-jet coaxial nozzle and Nd:YAG laser cladding system and cross-sections of the resulting coatings are presented in Figure 2. The cladding nozzle consisted of a central laser opening with inner shielding (with argon), an outlet opening for outer shielding gas and four circumferentially positioned outlet holes for the powder feed. A 2.7 kW diode-pumped Nd:YAG solid-state laser (Rofin) with a wavelength of λ = 1064 nm and 7-axis robot (ABB), located at the Bay Zoltán Research Institute (Budapest, Hungary), was used to deposit the laser cladding. Prior to the depositions, several trial tests were conducted to select the laser cladding parameters for cladding of the pure NiCrBSi powder: the power *P*, scanning speed *S* and feed rate $\dot{m}$. The laser transverse electromagnetic mode *TEM*_00_ was used with a Gaussian energy distribution through the laser spot. Nine parallel tracks were made with a length of 40 mm and with 40% of transverse overlapping. The thickness of the final coatings was between 900 and 1000 μm. For a continuous wave, Engström [38] and Wu et al. [39] introduced two combined parameters, the energy density (1) and power density (2), that are calculated as:(3)$$E=\frac{P}{S\cdot d}$$
(4)$$G=\frac{\dot{m}}{S\cdot d}$$
where *P* is the power (W), *S* is the scanning speed (mm/s), $\dot{m}$ is the feed rate (g/min) and *d* is the laser diameter on the surface of the substrate (mm). In order to establish reasonable initial starting laser cladding parameters for the various powder alloys, single track preliminary experiments were conducted over a range of powers (levels 1000, 1500, 2000 and 2500 W), scanning speeds S (levels 2, 4, 5, and 8 mm/s) and powder feed rates (10, 12 and 14 g/min) to find the optimal combination of parameters. The objective was to keep the geometrical dilution around 0.1 and achieve an effective deposition thickness above 0.5 mm. The geometrical dilution is a ratio between the depth of the melted substrate underneath the cladding and the sum of the effective deposition thickness and depth of the melted substrate [40]. The selected laser cladding parameters for multi-track overlapped deposits are given in Table 3.

### 2.3. Surface Roughness

The surface roughness after laser cladding was measured with a contact profilometer (model Dektak XT, Bruker, Karlsruhe, Germany) over the top of the cladded area with dimensions of 5 mm × 2 mm. This analysis was conducted to determine the surface conditions after laser cladding and before grinding and polishing for wear tests. The surface roughness parameter selected to describe the surface was the arithmetic mean height parameter *Sa*. This is a 3D parameter expanded from the roughness (2D) parameter *Ra*. It expresses the average of the absolute values of *Z*(*x*,*y*) in the measured area. It is equivalent to the arithmetic mean of the measured region on a three-dimensional display diagram when valleys were changed to peaks by conversion to absolute values. Mathematically, *S_a_* is defined as follows:(5)$$S_{a}=\frac{1}{A}\overset{\;}{\underset{a}{\iint }}\left|Z\left(x,y\right)\right|dxdy$$
where *Z*(*x*,*y*) is the function representing the height of the surface relative to the best fitting plane. The “a” used in the following integral expressions implies that the integration was performed over the measurement area and then normalized by the cross-sectional area “A” of the measurement. For the surface roughness measurement, the surfaces were ground with a sequence of SiC grit papers from 220 to 1000 grit and polished with 3 and 1 μm diamond pastes.

### 2.4. Microstructural Analysis with EDS/EBSD and XRD

X-ray diffraction (XRD) analyses were conducted using a PANalytical X’Pert PRO spectrometer (Malvern Panalytical, Malvern, UK) with a Cu anode that was operated at 40 kV and 45 mA. The spectra were recorded over a 2θ angular range from 25° to 95° with a step size of 0.002° and collection time of 30 s/step. The diffractograms were analyzed using HighScorePlus software (Malvern Panalytical, Malvern, UK).

A JEOL JSM 6500-F (JEOL, Tokyo, Japan), equipped with Oxford Instruments EDS (Oxford Instruments NanoAnalysis, High Wycombe, UK), a HKL Channel5 EBSD system (Oxford Instruments, Oxon, UK), Zeiss CrossBeam 550 EDAX EDS (Zeiss, Oberkochen, Germany) and EBSD (Zeiss, Oberkochen, Germany), was used for microstructural and phase analyses. Data processing and analysis were performed using the Channel 5, Inca, Team and TSL OIM software packages. The specimens for metallographic analysis were taken from transverse cross-sections that were ground and polished. For the EBSD analysis, final polishing with OP-S was performed for 5 min. The specimens were etched with a 10% Nital solution (substrate), and the NiCrBSi coating was etched with a 50% HCl solution, 33% glycerol and 16% HNO_3_.

### 2.5. Hardness and Micro-Hardness Testing

Measurements of the hardness across the cross-section of the coating were performed with an indenter (HV_0.2_) and a load of 1.9 N with a Leitz Wetzlar gage, Leica Microsystems (Leica, Wetzlar, Germany). Each measured point represents the average of 4 indentations at a constant depth below the surface. In addition, the hardnesses of the individual micro-constituents, such as the matrix and reinforced particles, were measured with a load of 0.0981 N (HV_0.01_).

### 2.6. Tribological Tests

Unlubricated tribological tests were conducted on a standard tribometer (CSM instruments, Anton Paar GmbH, Ostfildern-Scharnhausen, Germany) at room temperature using a reciprocating sliding ball on a disc with a counterpart spherical ball of Al_2_O_3_ (HV ≈ 2000) with a 6 mm diameter, Figure 3. The tests were performed at a frequency of 2 Hz, a 5 mm stroke and a normal constant load of 5 N, which led to a nominal initial Hertzian contact pressure of 1.3 GPa on the NiCrBSi coating (Al_2_O_3_; E_1_ = 380 GPa, ν_1_ = 0.3, NiCrBSi coating E_2_ = 210 GPa, ν_2_ = 0.3). Due to the high modulus of the WC-12Co composite (E_2_ = 210 GPa∙0.12 + 530 GPa∙0.88 = 490 GPa), the Hertzian contact pressure increased with the WC-12Co content, with a maximum in the 100%WC-12Co coating of 2.5 GPa. The tribological parameters were selected to cover a relatively large hardness range of different investigated materials. After 2000 cycles, the total wear length was 10 m. Before testing, all deposited coatings were levelled and polished. The environmental conditions were kept constant (T = 23 ± 2 °C and RH = 50 ± 10%). The wear tracks were examined using light microscopy, and the friction and wear properties were compared and correlated with the WC-Co content and the resulting microstructures. The sample wear rate was assessed by measuring the wear volume using a Bruker Dektak XT, where the measurement length was 3 mm, and the tip radius of the diamond needle was 2 μm. Three tests were conducted for each wear test condition, and the average value is given. The counter-body (Al_2_O_3_ sphere) worn area was measured using a Zeiss Axio CSM 700 with a 10× objective and 150× magnification.

## 3. Results and Discussion

### 3.1. Surface Roughness after Laser Cladding

The laser cladding parameters, such as the power, scanning speed, powder feed rate, overlap and movement trajectory, were kept constant during the deposition, as shown in Table 3. Therefore, the surface topography and resulting *S_a_* roughness varied with the chemical composition of the powder, as shown in Figure 4. The laser beam reflectivity and absorption with the powder mixture and surface varied depending on the powder chemical composition [41,42,43]. This included a large difference in the melting temperature, where the NiCrBSi had a liquidus point at 1180 °C, and the WC had a melting temperature of 2850 °C. There were also differences in the thermal conductivities; for example, the thermal conductivity of WC-Co was approximately twice that of nickel. 

The NiCrBSi surface was relatively smooth. Three were peaks from adhered powder particles at the top that increased the overall surface roughness, leading to a surface roughness of *S**_a_* = 11.7 μm, as shown in Figure 4a. The adhered particles were mostly unmelted or partially melted particles that escaped melting in the core of the laser beam.

In the NiCrBSi coatings with 30, 40 or 50 wt.% of WC-12Co, the *S_a_* roughness reduced to values from 5.5–7.8 μm, as shown in Figure 4b–d. The visual appearance of the NiCrBSi coating was light metallic, whereas the appearance of the NiCrBSi with 30, 40, 50 wt.% of WC-12Co and 100 wt.% WC-12Co was darker due to the dark colour of the carbides. The highest roughness was on the WC-12Co coating with *S_a_* = 12.3 μm, Figure 4e. Prior to the wear tests, the cladding surfaces were levelled using a diamond mill and subsequently finely ground and polished to *Ra* = 0.01–0.05 μm. 

### 3.2. Microstructures after Laser Cladding

Backscattered electron images of the coatings are shown in Figure 5. Panel (a) represents the microstructure of the laser cladded NiCrBSi coating, where the main observed features consisted of planar cellular-dendritic γ-Ni and Ni/Ni_3_B eutectic. The EDS analysis of the different surface areas is reported in Table 4, with two to four repeated measurements for each value. The matrix phase region of the NiCrBSi coating, spectrum 1, contained 80.7 wt.% of Ni, 12.2 wt.% of Cr and 7.1 wt.% of Si. The fine-grained eutectic γ-Ni /Ni_3_B zone, spectrum 2, contained 78.1% Ni, 12.6% Cr, 6.7% Si and 2.6 wt.% O. Moreover, influence of WC-12Co content on the coating and matrix hardness and (b) variation of the main alloying elements are shown in Figure 6. The dark etched areas (spectrum 3) were rich in Cr. To confirm the individual constituents of the microstructures, we also performed EBSD point analysis on the different phases and identified M_7_C_3_ carbides and CrB-type borides. Representative Kikuchi patterns are shown in the image. The XRD analysis, reported in Figure 7b for all coatings, confirmed the presence of the Ni/Ni_3_B phase, Cr_7_C_3_ carbide and CrB boride. The XRD spectrum of the coating shows the presence of the Ni/Ni_3_B phase, Cr_7_C_3_ and CrB. The CrB borides and Cr_7_C_3_ carbides are often present in NiCrBSi coatings, which was also confirmed by several authors [19,44].

The microstructures of the mixed coatings contained several additional phases. The 70% NiCrBSi/30 wt.% WC-Co consisted of a matrix phase (dark grey) that was surrounded by brighter hard phases, as shown in Figure 5b. The EDS data implied that the dark grey area of the matrix was alloyed with Co (3.4 wt.%) and W (6.2 wt.%) and formed a multi-element NiCrCoW solid solution. Due to the alloying, the NiCrCoW matrix had a hardness that was 1.4-times harder that of the original γ-Ni-matrix in the NiCrBSi coating. To alloy the matrix, the cobalt was first detached from the WC-12Co binder (with 12% cobalt), whereas the tungsten came into the matrix after decomposition of the WC. Decomposition was followed by the mixing and diffusion of W and Co in the melt, which was then incorporated into the solid matrix solution. Similarly, Niranatlumpong et al. [34] reported the formation of a NiCrW solid solution after arc-spraying but to a lesser extent; as a result of the rapid solidification of splats, the time for diffusion was short. The Ni and Co were soluble (α-Co, α-Ni) throughout the entire range of concentrations [45]. Alloying Ni with Co and W increased the hardness and toughness of the matrix and wear resistance. The matrix hardness increased from 430 to 601 HV_0.01_. The hardness of the coating and matrix phase in the coating exponentially increased with increasing WC-12Co content, as shown in Figure 6a. Figure 6b show the plotted EDS data for the evolution of the Ni, Cr, Co and W in the NiCrCoW solid solution. The data revealed that the Co and W weight contents increased while the Ni and Cr weight contents in the matrix phase decreased with increasing WC-12Co.

Another constituent in the microstructure of the mixed coatings was undissolved tungsten carbides (WC), which were the hardest particles in the microstructure (2527 HV_0.01_). As a result of the WC decomposition, graphite particles were sparsely distributed. In the NiCrBSi with 30, 40 and 50 wt.% WC-12Co, the EBSD results confirmed the presence of WC, Cr_7_C3, Cr_23_C_6_, γ-Ni and Ni_3_B, as shown in Figure 7. Occasionally, larger WC particles became dissolved or thermally cracked during heating and cooling, as shown in spectrum 3. The light etched particles had a high oxygen content from 16–25 wt.% with a high percentage of Ni, W and Fe and hardness of 1467 HV_0.01_.

The microstructure of NiCrBSi + 40 wt.% WC-Co, Figure 5c, consisted of the light grey NiCrCoW matrix with a hardness of 714 HV_0.01_. Occasionally, elongated oxides were present with a hardness of approx. 1125 HV_0.01_. The NiCoCrW matrix represented 60 % of the area fraction, followed by 27% WC and the rest with oxides. Compared to that for previous samples, the WC carbides were smaller in size.

The microstructure of the NiCrBSi + 50 wt.% WC-Co contained smaller WC and oxide particles, Figure 5d. The NiCoCrW matrix represents 50% of the area fraction, followed by WC with 32% and the rest with oxides. The hardness of the NiCoCrW matrix increased to 747 HV_0.01_.

To identify the new phases that formed during laser cladding, XRD analysis was performed on the initial powders, and the results are shown in Figure 7a. The WC-12Co powders show only reflections belonging to WC and Co, and Rietveld refinement showed a good agreement with the overall composition (11% Co and 89% WC). The diffractogram of the NiCrBSi and mixed powders were more complex, and a number of peaks belonging to different phases can be seen.

The main peaks belong to the WC and Ni/Ni_3_Si phases, while there are a large number of peaks with very low intensity. The best candidates for these smaller reflections were identified to be Ni_3_B, Cr_7_C_3_, and Cr_23_C_6_ based on the known chemical composition of the sample. The peaks are less than 1% intensity with regards to the largest peak; therefore, the accurate identification of these phases is not possible.

X-ray diffractograms of the mixed coatings are shown in Figure 7b. The two main constituents were WC and Ni/Ni_3_Si phases, and additional minor peaks were also observed that belong to the CrB and Cr-carbide phases. The microstructure of the WC-12Co coating contained three main components, as shown in Figure 5e. The matrix phase was Co-rich (β), and there were two carbide types: large WC carbides that were polygonal with sharp edges and a network of smaller carbides that had a different Kikuchi pattern that corresponded well to the M_6_C (Co_3_W_3_C) phase. The XRD measurements of the coating confirmed the presence of WC, W_2_C, Co_3_W_3_C and the matrix Co-rich phase.

### 3.3. Through-Depth Hardness Measurements

Figure 8 show the through-depth hardness measurements (N = 4 for each depth) for the tested coatings. The NiCrBSi had an average hardness of 573 HV_0.2_, followed by NiCrBSi with 30 wt.% of WC-12Co with an average hardness of 730 HV_0.2_, 40 wt.% of WC-Co had an average hardness of 838 HV_0.2_, 50 wt.% WC-Co had an average hardness of 1067 HV_0.2_ and WC-CO had the highest average hardness of 1695 HV_0.2_. By increasing the WC-Co content, the scatter in the hardness increased due to the high difference in the hardness between the matrix and WC reinforcing particles. Due to the absence of hard WC particles in the 100% NiCrBSi coating, the hardness of the coating did not vary significantly. The variation of hardness is the largest in 100%WC-12Co coatings. The main source of variation is the heterogeneity of the material tested. The addition of WC-carbides increased the macro-heterogeneity and the hardness variation of the material. For example, in WC-12Co coatings, there are relatively large WC-carbides present with the size of over 10 μm. The indentations that were located partly or directly on WC-crystal yielded to high hardness; the maximum measured value with a load of F = 1.96 N was 2004 HV_0.2_. When the indentation was located in a region of much softer Co-matrix with dendrites, the minimum value was 1468 HV_0.2_.

### 3.4. Wear Properties

The profiles after wear for three repeated tracks are shown in Figure 9. The NiCrBSi profile had a semi-elliptical shape with the deepest worn-out cross-section and area. The profiles for the NiCrBSi with 30, 40 and 50 wt.% WC-12Co were very shallow with a significantly smaller area, indicating a large increase in the wear resistance. The worn profiles for the NiCrBSi-30 and 50 wt.% WC-Co coatings had sharp asperities at the bottom surface below the counter-body after the wear test, which was due to the removal of the softer matrix and detaching of the hard carbides and adhered debris around the groove. In the case of the WC-12Co coating, the profile was only 0.1–0.2 μm deep, and the wear debris had a larger area than the worn-out areas.

In Figure 10, the coefficient of friction (COF) values are plotted against the number of cycles for the NiCrBSi/Al_2_O_3_ and WC-Co/Al_2_O_3_. The COF for the NiCrBSi coating was the highest at the beginning of the test (*μ* = 0.64), followed by steady-state wear with a COF from 0.52–0.59. The high wear rate of the NiCrBSi, deep grooves in the wear tracks and SEM micrographs revealed severe wear damage due to the adhesive wear. During adhesive wear, the typical events include continuous formation and breaking of interfacial adhesive bonds, e.g., cold-welded junctions. After being welded, sliding forces tear the metal from the surface. The result is a microscopic cavity on one surface and a projection on the other. Adhesive wear initiates microscopically but progresses macroscopically [46]. The formation and breaking of bonds could also be the reason for the COF variation. The large wear rate was accompanied by the NiCrBSi debris that adhered to the edges of the worn surface of the Al_2_O_3_ counter-body, as shown in Figure 11a.

The lowest average COF was found for the WC-12Co/Al_2_O_3_ pair, with a starting value of 0.12. It increased rapidly to a value of 0.15, and then it slowly increased to 0.17 until 770 cycles were reached. The fine darker wear debris formed, creating three-body abrasion wear. The friction gradually increased after 770 cycles to 0.37 and after 1700 cycles to 0.52. This indicated the gradual formation of rougher micro-asperities on the surface.

The friction coefficients in the NiCrBSi + 30, 40, 50 wt.% WC-12Co coatings were also very low (0.15) at the beginning. Then, they increased rapidly with large up-down oscillations within 750 cycles for the 30 wt.% and 40 wt.% WC-12Co cases and 250 cycles for the 50 wt.% WC-12Co, as shown in Figure 10b. In this running-in phase, the combination of adhesive and three-body abrasive wear occurred until the removal of the surface layer. The adhesion wear prevailed until the ball finally removed the binder layer that covered the carbides, and afterwards, the ball started to slide more on the top of the carbides. In coatings with a higher percentage of WC-12Co, the oscillations ended earlier since there was less binder to be removed before the steady-state was reached.

In Figure 10b, after the initial oscillations, the COF reduced with an increase in the WC-12Co content. In the case of the coating with 50 wt.% WC/Co, the coefficient of friction decreased after 1250 cycles from 0.53 to 0.42 at the end of the test. The variation of the friction coefficient can be related to the formation of smoother surfaces after sliding or adhesive wear removal of the softer binder.

The optical microscopy images of the wear tracks from the Al_2_O_3_ counter-body are shown in Figure 11. In the first image, the NiCrBSi large debris flakes at the edges of the wear track adhered to the ball. As a result, the formation and breakage of micro cold-welded junctions led to the detachment of Al_2_O_3_ particles in contact with the surface and concentrated at the centre of the wear track, as shown in Figure 11a. For all other cases of sliding wear against NiCrBSi with 30, 40, 50 wt.% WC-12Co and WC-12Co, a similar case of adherence can be found but with a fine black powder; the wear of the oxide counter-body for the 30, 40 and 50 wt.% WC-12Co coatings are shown in Figure 11b–d. In the case of coatings with pure WC-Co, only a small amount of debris can be found on the surface with micro-cutting traces in the sliding direction, as shown in Figure 11e.

The corresponding wear rate was calculated using an Archard equation [47]:(6)$$w_{e}=\frac{V}{F_{N}\cdot l}\;\;\;\;$$
where *V* is the volume of wear track, *F*_N_ is normal load and *l* is the length of one pass (5 mm). The volume of the wear track was calculated as:(7)$$V=A\cdot l_{t}$$
where *A* is the area of wear track and *l_t_* is the total length of wear.

In Figure 12, a histogram for the wear rate with the average coefficient of friction is shown. The average coefficient of friction decreased with increasing WC-Co in the coating. The radius of the wear in the Al_2_O_3_ counter-body decreased with increasing WC-Co content. The wear rate of the NiCrBSi coating was significantly higher than the wear rates of the other samples with WC-Co because of the adhesion type of wear. 

#### SEM Wear Track Analysis

The SEM images of the worn surfaces are shown in Figure 13. On the laser cladded NiCrBSi coatings, adhesive wear mode with micro-ploughing and micro-cutting marks are visible, as shown in Figure 13a. The coating material was removed repeatedly along with the detached adhesive material. To a smaller extent, detached abrasive small debris was formed and led to three-body abrasive wear. Debris accumulated in delaminated holes with a high percentage of Al and O (Al_2_O_3_ counter-body debris) and Cr-rich particles. In additon, surface fatigue was present in the form of repeated sliding, forming maximum repeated shear stresses below the surface. Smaller metallic wear debris was oxidized.

The wear track in the laser cladded NiCrBSi with 30 wt.% WC-Co is shown in Figure 13b. In this case, the main wear mechanism was abrasion. When particles became detached, the small hard particles caused abrasive micro-cutting wear. Debris was formed during the first 500 cycles of sliding wear. After the binder material was removed, the WC particles detached, and debris accumulated around the wear tracks. The surface fatigue can be characterized by crack formation in the matrix around the WC and crushed WC. Furthermore, cracks were found at the matrix-WC grain boundaries. Localized fatigue occurred on a microscopic scale due to repeated sliding contact of the asperities on the surfaces of the solids in relative motion. Fatigue cracks originated around the sharp edges on the WC and also formed below the hard particles. The fatigue of materials proceeds in the sequence of elastic and then plastic deformation, work hardening or work softening, crack initiation and crack propagation. The formation and propagation of cracks due to repeated loading can also result in shallow pits. The critical contact shear stresses also impacted the fatigue and reached a maximum value below the surface. The maximum Hertzian contact shear stress increased by internal notches; WC particles caused subsurface cracking, for example. Cracking was due to a local high pressure at the edges of the WC particles that acted on the matrix.

Surface fatigue can also be important during sliding contact between two solids. Similar to the mechanism of adhesion or abrasion, repeated sliding of hard asperities across a solid surface can cause crack formation and crack propagation at or below the stressed surface. According to the model proposed by Suh and co-workers [48], subsurface cracks propagate parallel to the surface. Wear particles are generated when a subsurface crack breaks through to the surface. In general, this surface disintegration by delamination is due to the simultaneous action of adhesion and/or abrasion and surface fatigue. The cyclic loading of a stressed surface results from friction and normal forces in the area of the contact. Friction forces arise from adhesion and/or abrasion between the asperities and the wearing surface. Similarly, as for the NiCrBSi without WC, there were occasional regions of delaminated Ni-Co-W matrix around the hard WC particles. Figure 13c show a wear track in the NiCrBSi with 40 wt.% WC-Co. A large number of WC particles shielded the matrix from adhesive wear delamination and micro-cracking. Larger angular WC blocks cracked during reciprocal sliding wear.

During sliding, WC particles detached from the surface, causing small wear groves in the matrix. Figure 13d show a wear track in NiCrBSi with 50 wt.% WC-Co, where the wear of the coating reduced, but the wear of the Al_2_O_3_ ball increased. On the hard WC particles, Al_2_O_3_ accumulated, confirmed by the high Al and O content at the surface of the WC blocks. Figure 13e show a wear track in the WC-Co coating. Detached WC particles from the Co matrix can be found. The wear groves were small, and light debris from the Al_2_O_3_ counterpart is visible.

## 4. Conclusions

Laser cladding of NiCrBSi with WC-12Co resulted in a multi-matrix highly wear-resistant coating. The primary self-fluxing binder, NiCrBSi, was solid strengthened with cobalt and tungsten from agglomerated the WC-12Co powder. The interaction of alloying elements and the addition of hard WC particles significantly reduced the wear rates. Based on the results herein, the following conclusions can be drawn:

The NiCrBSi coating consisted of planar cellular-dendritic γ-Ni and Ni/Ni_3_B eutectic. EBSD point analysis of the different phases revealed M_7_C_3_ (Cr_7_C_3_) carbides and CrB-type borides. Despite the relatively high hardness of approximately 430 HV_0.2_, the coating suffered severe adhesion wear and high wear rates during reciprocal sliding of the Al_2_O_3_ ball. The intensive adhesion wear with deep wear tracks and the highest coefficient of friction was observed in the NiCrBSi coatings. Notable amounts of debris and oxides of NiCrBSi were found attached to the Al_2_O_3_ counterpart ball.
High-energy laser cladding of NiCrBSi with WC-12Co enabled the cobalt and tungsten to be dissolved and integrated into the matrix to form a multi-matrix NiCoCrW The addition of 30, 40, 50 wt.% WC-Co and resulting alloyed nickel-based matrix with Co and W resulted in 50,78 and 85% higher hardness values, respectively. In the NiCrBSi with 30, 40 and 50 wt.% WC-12Co, EBSD studies confirmed the presence of WC, Cr_7_C_3_, Cr_23_C_6_, γ-Ni and Ni_3_B. Occasionally, larger WC particles were dissolved or thermally cracked during heating and cooling. Alloying of the Ni-matrix with Co and W reduced the plastic deformation and differences in the hardnesses and prevented severe material removal by adhesive wear, which was present in NiCrBSi and Al_2_O_3_ pair.The abrasive wear mode was observed on the NiCrBSi coatings with 30, 40 and 50 wt.% WC-12Co. Compared to that for the original NiCrBSi laser cladded coating, the wear rates dropped by approximately 4.7 times for the 30 wt.% WC-12Co, 5.9-times for the 40 wt.% WC-12Co and a 13.5-times for the 50 wt.% WC-12Co. Thus, the wear rate decreased with increasing content of WC-12Co. In the NiCrBSi with 30 wt.% of WC-Co, wear fatigue cracks were found in the matrix between the WC-particles, and there was a higher amount of detached particles. In the NiCrBSi with 40 and 50 wt.% WC-12Co, wear fatigue cracks were not detected.The microstructure of the WC-12Co consisted of a Co-rich (β) phase and two carbide types: large polygonal WC carbides with sharp edges and a network of an M_6_C (Co_3_W_3_C) phase, WC and W_2_C. The WC-12 Co coating was the most difficult to laser clad because it was brittle and had the highest wear resistance among the tested combinations. Compared to that for the original NiCrBSi laser cladded coating, the wear rates of the WC-12Co coatings were approximately 23.1-times lower.

It was demonstrated that the NiCrBSi coating in a pair with Al_2_O_3_ was subjected to severe adhesion, much more than was expected. The combination of WC-Co with the self-fluxing NiCrBSi alloy yielded coatings with significantly higher wear resistance. Despite WC being blended with Co and deposited with a Ni-based alloy, the dissolution of WC and gaseous porosity in the NiCrBSi with WC-Co remained a problem. Therefore, laser cladding parameters must be adapted to the deposited material to prevent excessive heat during NiCrBSi + WC-Co deposition.

## Figures and Tables

**Figure 1 materials-15-00342-f001:**
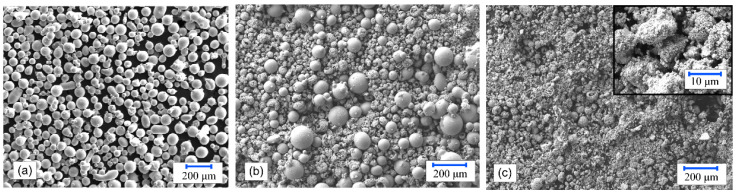
Powders used herein: (**a**) NiCrBSi, (**b**) NiCrBSi + 30 wt.% WC-12Co and (**c**) WC-12Co.

**Figure 2 materials-15-00342-f002:**
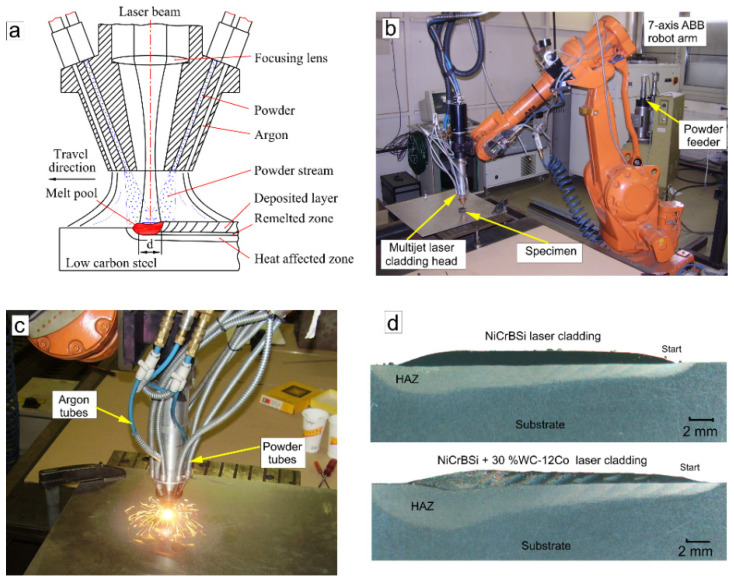
(**a**) Schematic of coaxial laser cladding head cross-section: (**b**) laser cladding system, (**c**) during laser cladding and (**d**) cross-section of the macrostructures.

**Figure 3 materials-15-00342-f003:**
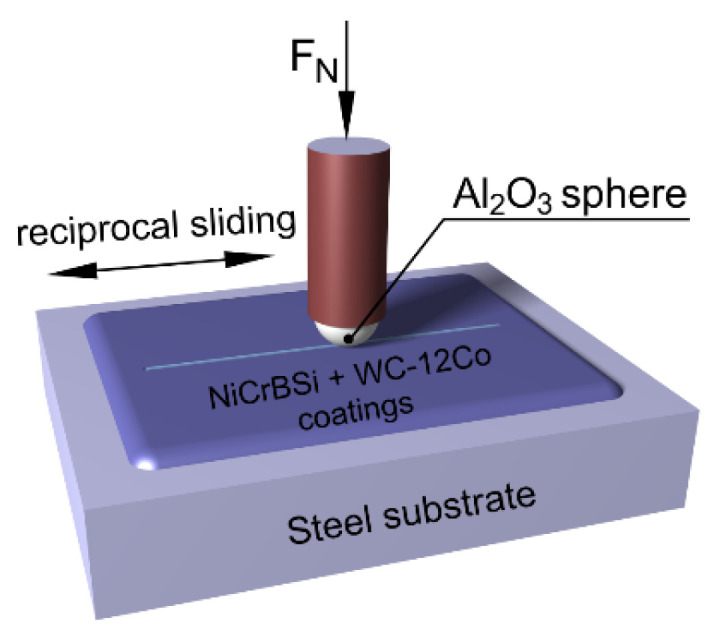
Scheme of the tribological test.

**Figure 4 materials-15-00342-f004:**
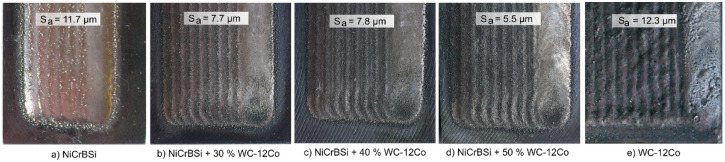
Surface roughness after laser cladding.

**Figure 5 materials-15-00342-f005:**
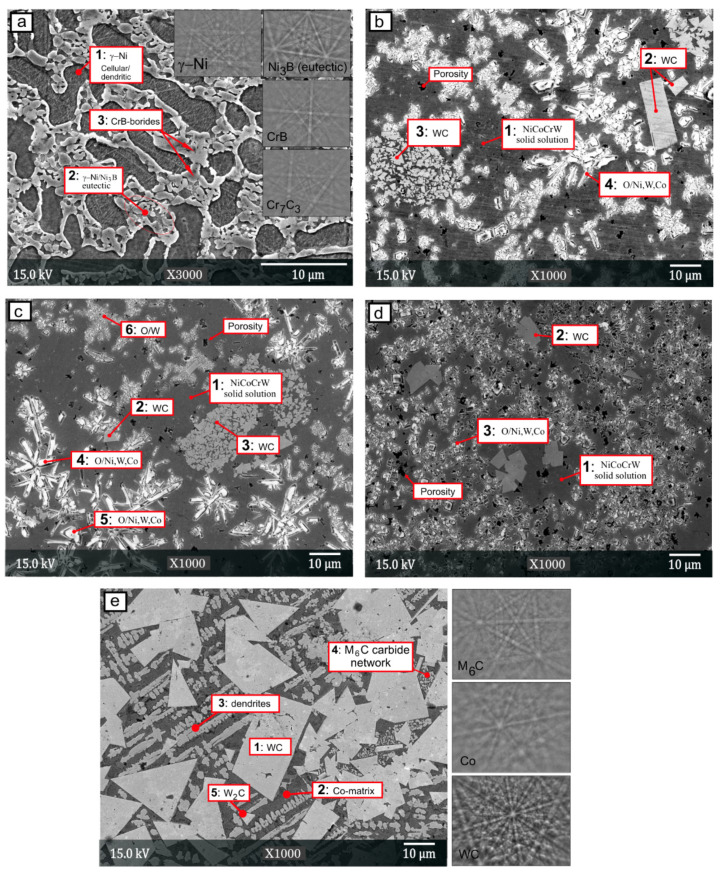
Microstructures of laser cladded coatings and Kikuchi patterns: (**a**) NiCrBSi, (**b**) NiCrBSi + 30 wt.% WC-12Co, (**c**) NiCrBSi + 40 wt.% WC-12Co, (**d**) NiCrBSi + 50 wt.% WC-12Co and (**e**) WC-12Co.

**Figure 6 materials-15-00342-f006:**
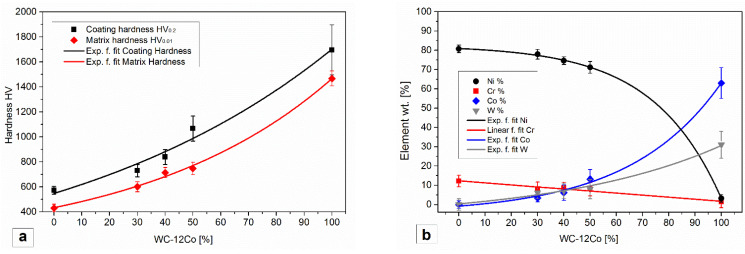
(**a**) Influence of WC-12Co content on the coating and matrix hardness and (**b**) variation of the main alloying elements.

**Figure 7 materials-15-00342-f007:**
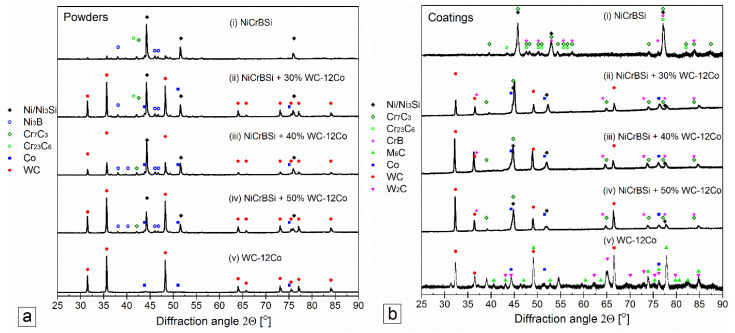
X-ray diffractograms for the (**a**) powders and (**b**) laser cladded coatings.

**Figure 8 materials-15-00342-f008:**
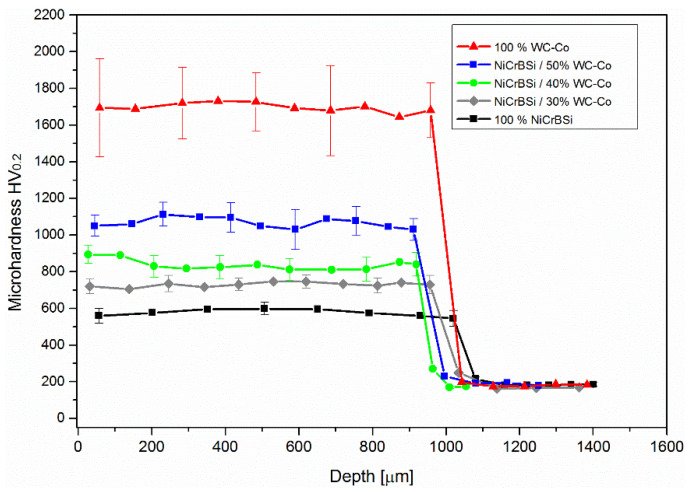
Through-depth microhardness.

**Figure 9 materials-15-00342-f009:**
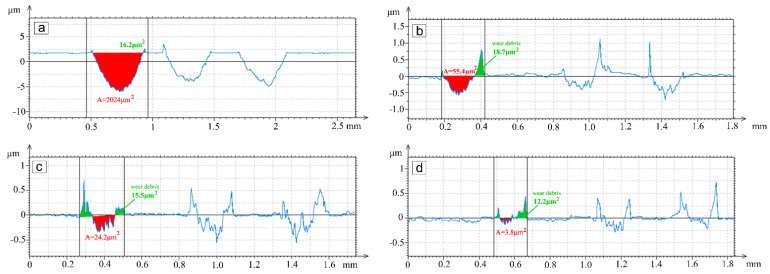
Depth of wear tracks from profile measurements: (**a**) NiCrBSi, (**b**) NiCrBSi + 30 wt.% WC-12Co, (**c**) NiCrBSi + 50 wt.% WC-12Co, (**d**) WC-12Co coating.

**Figure 10 materials-15-00342-f010:**
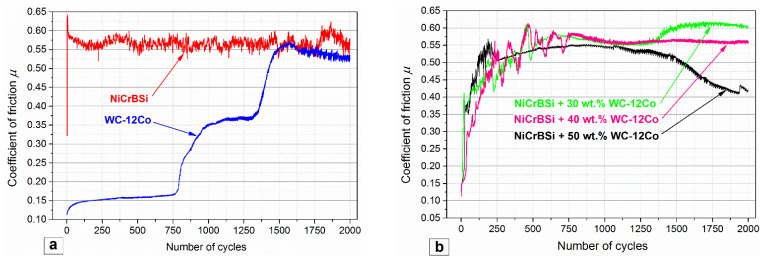
Coefficient of friction for the: (**a**) NiCrBSi and WC-Co and (**b**) NiCrBSi + 30, 40, 50 wt.% WC-12Co.

**Figure 11 materials-15-00342-f011:**
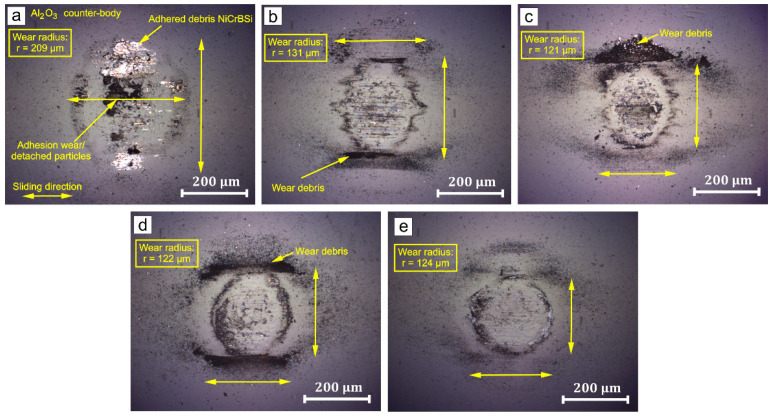
Wear of Al_2_O_3_ spherical counter-body on: (**a**) NiCrBSi, (**b**) NiCrBSi + 30 wt.% WC-12Co, (**c**) NiCrBSi + 40 wt.% WC-12Co, (**d**) NiCrBSi + 50 wt.% WC-12Co and (**e**) WC-12Co.

**Figure 12 materials-15-00342-f012:**
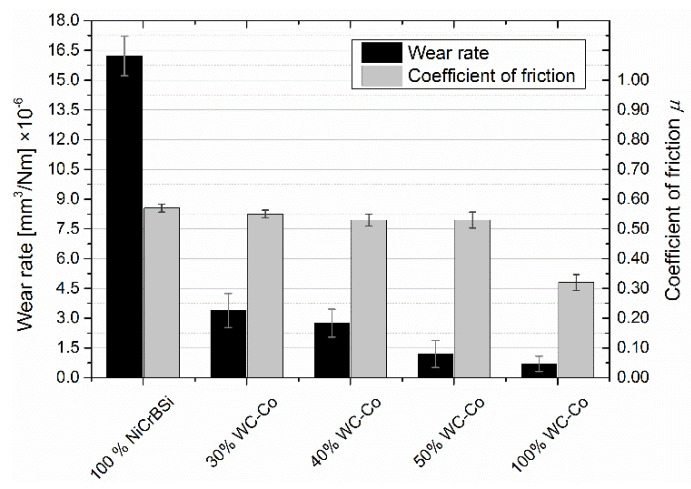
Histogram of the wear rate and friction.

**Figure 13 materials-15-00342-f013:**
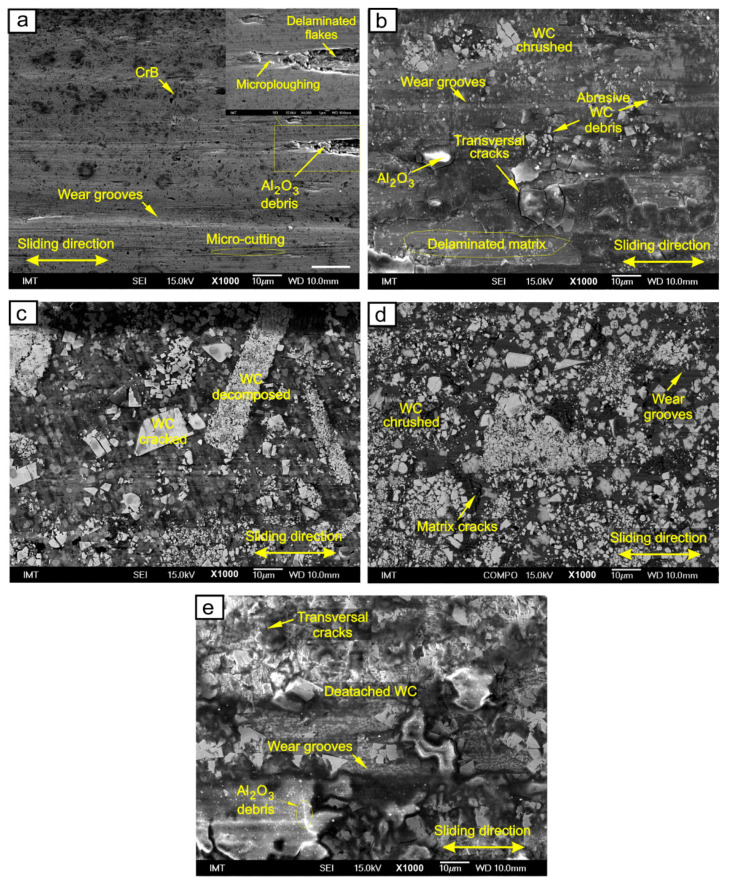
SEM wear tracks after 30 min of reciprocal sliding with F = 5 N: (**a**) NiCrBSi, (**b**) NiCrBSi + 30 wt.% WC-12Co, (**c**) NiCrBSi + 40 wt.% WC-12Co, (**d**) NiCrBSi + 50 wt.% WC-12Co and (**e**) WC-12Co.

**Table 1 materials-15-00342-t001:** Chemical compositions of feedstock materials.

Code	Comercial Name	Elements (wt.%)			
		Ni	Cr	B	Si
NiCrBSi	Eutalloy RW 12495	78	13	2.08	4
		WC		Co	
WC-12Co	SM 5810	88		12	

**Table 2 materials-15-00342-t002:** Powders.

Code	Powder	NiCrBSi (wt.%)	WC-12Co (wt.%)	Particle Size (μm)	Density (g/cm^3^)
NiCrBSi	Eutalloy 12,495	100	-	72	8.2
+30 wt.% WC-12Co	Exp 1	70	30	63–72	10.2
+40 wt.% WC-12Co	Exp 2	60	40	63–72	10.8
+50 wt.% WC-12Co	Exp 3	50	50	63–72	11.5
WC-12 Co	SM 5810	-	100	−63 + 5	14.8

**Table 3 materials-15-00342-t003:** Solid-state Nd:YAG laser cladding parameters.

Laser Power *P* (W)	Scanning Speed *S* (mm/s)	Powder Feed $\dot{m}$	Laser Diameter *d* (mm)	Overlap (%)	Energy Density *E* (J/mm^2^)	Powder Density *G* (g/dm^2^)
2000	8	10	2.67	40	94	78

**Table 4 materials-15-00342-t004:** Chemical composition (EDS) and Vickers microhardness of NiCrBSi + 30, 40, 50 wt.% WC-12Co claddings at different points.

Coating Material	Spectrum POSITION	Name	Hardness HV_0.01_	Chemical Composition (wt.%)					
				Ni	Cr	Si	W	Co	O
NiCrBSi	1	γ-Ni	430	80.7	12.2	7.1	0	0	0
	2	Eutectic	750	78.1	12.6	6.7	0	0	2.6
	3	Cr-rich	763	76.5	18.1	3.9	0	0	1.5
NiCrBSi + 30 wt.% WC-12Co	1	NiCrCoW	601	77.9	8.1	4.4	6.2	3.4	0
	2	WC	2527	1.1	0	0	98.0	0	0.9
	3	Crushed WC	1620	18.2	2.7	0	77.1	1.1	0.9
	4	Oxides	1467	31.5	5.4	0	28	3.1	32
NiCrBSi + 40 wt.% WC-12Co	1	NiCrCoW	714	74.6	8.6	3.5	7.1	6.2	0
	2	WC	2463	2.1	0	0	97.9	0	0
	3	Crushed WC	2089	10.9	1.4	0	85.7	0.9	1.1
	4	Oxides #1	1125	31.4	2.8	1.6	6.9	3.8	53.5
	5	Oxides #2	*	36.0	3.6	1.2	7.3	5.2	46.7
	6	Oxides #3	*	15.1	7.2	0	63.6	2.1	12
NiCrBSi + 50 wt.% WC-12Co	1	NiCrCoW	747	71.1	7.8	0	7.90	13.2	0
	2	WC	2533	0	0	0	100	0	0
	3	Oxides	1159	22.3	5.3	0	48.5	6.4	17.5
	4	Oxides	*	35.8	4.7	0	41.1	0	18.4
WC-12Co	1	Co-matrix	1467	3.2	1.4	0	31.4	63.4	0
	2	WC	2281	0	0	0	100	0	0
	3	Dendrites	*	0	0	0	88	12	0
	4	M_6_C carbide network	*	0	0	0	56	44	0

* Measurements were not performed.

## Data Availability

Data is contained within the article.

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
