# Peer review of "Tribological Properties of Solid Solution Strengthened Laser Cladded NiCrBSi/WC-12Co Metal Matrix Composite Coatings"

_materials, 2022, doi:10.3390/ma15010342_

Round 1

Reviewer 1 Report

The subject matter is very interesting, important, and has a special value considering practical applications. The paper is clearly presented and well organized. The authors have carried out a thorough literature review which justified their research. References are adequate considering the discussion in the paper. The research methods and materials have been carefully characterized. Although I would propose to present a tribological research scheme. The authors have used modern research methods in their research, such as EBSD.
I suggest a mandatory revision of the following points to increase the quality of the paper: 
1. Please explain the selection of the parameters of the tribological test.
2. In my opinion, the authors should investigate the hardness of coatings on the surface with a higher load, i.e. 1 kgf.
3. They should insert the tribotest scheme.

Recommendation: 
This manuscript in the presented form is not acceptable for publication in the Materials. The minor revision is necessary.

Reviewer 2 Report

The manuscript entitled Tribological properties of solid solution strengthened laser cladded NiCrBSi/WC-12Co metall matrix composite coatings describes the obtaining process for several compositions and the involved parameters of multi-jet coaxial laxer cladding with Nd:YAG.
The introduction is logically written and clearly exposed on a solid base of references. In fact, each part of the manuscript was carefully constructed and extensively and deeply described.
Although, I have no doubt regarding the scientific level of the presented work, the last paragraph of the introduction should be reconstructed in such way to expose more clearly and specifically the novelty of this manuscript, compared to the previous researches presented in the introduction.  The authors presented the scientific background, advantages and drawbacks of various methods, sensible aspects of the research and so on. Then in the last paragraph they state what the present manuscript deals with. Therefore, as I mentioned before, a phase that highlights the novelty should be added.
Regarding the materials and methods, the substrate materials, feedstock materials, laser cladding process and all characterization methods are properly described.
I appreciated mostly the results and discussion chapter, clearly,  detailed, with suggestive graphs and microstructures images.
Considering the overall merit of the authors I cannot wonder how come an fast and simple characterization method like the hardness test was presented as the average of only 4 indentations (page 6, 243). Maybe this is not a problem, BUT I need a clarification on Figure 7: for 100%WC-Co the error bars range from 1400 to 2000 HV? If it is true, you could consider increasing the number of indentations....that is a large range...

As a personal curiosity, could you mention the model and year of the Taylor-Hobson profilometer? I use a similar one and I don't have the Sa roughness feature.

In conclusion, I can only appreciate the amount of work presented by the authors and as the two minor aspects will be addressed by the authors the manuscript can be published with minor changes.
